# Phosphonic Acid Analogues of Phenylglycine as Inhibitors of Aminopeptidases: Comparison of Porcine Aminopeptidase N, Bovine Leucine Aminopeptidase, Tomato Acidic Leucine Aminopeptidase and Aminopeptidase from Barley Seeds

**DOI:** 10.3390/ph12030139

**Published:** 2019-09-17

**Authors:** Weronika Wanat, Michał Talma, Małgorzata Pawełczak, Paweł Kafarski

**Affiliations:** 1Department of Bioorganic Chemistry, Wrocław University of Science and Technology, Wybrzeże Wyspiańskiego 27, 50-370 Wrocław, Poland; weronika.wanat@pwr.edu.pl (W.W.); michal.talma@pwr.edu.pl (M.T.); 2Faculty of Chemistry, University of Opole, ul. Oleska 48, 45-052 Opole, Poland; Malgorzata.Pawelczak@uni.opole.pl

**Keywords:** aminopeptidases, inhibitors, aminophosphonate, phenylglycine analogues, fluorine substituted, molecular modeling

## Abstract

The inhibitory activity of 14 racemic phosphonic acid analogs of phenylglycine, substituted in aromatic rings, towards porcine aminopeptidase N (pAPN) and barley seed aminopeptidase was determined experimentally. The obtained patterns of the inhibitory activity against the two enzymes were similar. The obtained data served as a basis for studying the binding modes of these inhibitors by pAPN using molecular modeling. It was found that their aminophosphonate fragments were bound in a highly uniform manner and that the difference in their affinities most likely resulted from the mode of substitution of their phenyl rings. The obtained binding modes towards pAPN were compared, with these predicted for bovine lens leucine aminopeptidase (blLAP) and tomato acidic leucine aminopeptidase (tLAPA). The performed studies indicated that the binding manner of the phenylglycine analogs to biLAP and tLAPA are significantly similar and differ slightly from that predicted for pAPN.

## 1. Introduction

Aminopeptidases are ubiquitous enzymes, which are widely distributed throughout the all biological kingdoms and found in many subcellular organelles, in the cytoplasm, and as membrane components. They constitute a group of exopeptidases with the capacity to cleave peptide bonds from the N-terminus of protein substrates [1,2] and play central roles in protein turnover, protein maturation and generation, or in the catabolism of bioactive peptides, which are important in a variety of physiological processes [1,3]. Most importantly, human aminopeptidases are considered as important targets in the search for potential anti-cancer agents [2,4,5,6], and the synthesis and evaluation of their inhibitors is an extensively developing field of research [6,7,8,9,10,11].

It is well known that libraries of structurally related, low molecular-weight enzyme inhibitors are a good starting point for the design of more potent, structurally more complex inhibitors, following the analysis of their interactions with enzymes at the molecular level [12]. Quite frequently, molecular modeling is of use as a tool in such studies. Having a series of phosphonic acid analogues of phenylglycine [13] variably substituted in aromatic rings in hand, we decided to determine their activity towards the commonly studied aminopeptidase N (alanine aminopeptidase, pAPN) from porcine kidney [11,14,15], which is a good model enzyme for human alanine aminopeptidase (hAPN), because the two enzymes contain one zinc ion in the active site and exhibit a similar pattern of activity towards series of the same synthetic substrates [16]. The human enzyme is overexpressed in various types of cancer and on the surface of the vasculature undergoing angiogenesis and is thus considered as a promising target for anticancer therapy [4,10,11,12,13,14,15,16,17]. The pattern of inhibitory activity found for pAPN was compared with the experimentally determined activity of the same set of compounds towards the newly isolated and only preliminarily characterized aminopeptidase from barley seeds [18,19].

In order to better understand the modes of binding of these inhibitors, molecular modeling was performed for pAPN. Our results indicate the different manner of interaction of single phosphonic acid analogues of phenylglycine with the pAPN spacious hydrophobic site, which normally binds aromatic portions of these molecules, whereas the binding of their aminophosphonate fragment is practically uniform.

For comparison, the mode of the possible binding of the same set of inhibitors to leucine aminopeptidase from bovine lens (blLAP) was modeled and compared with data obtained for tomato acidic leucine aminopeptidase (tLAPA), the only plant enzyme for which a crystal structure has been described to date [20]. These studies have been done in order to determine the similarity between these two enzymes, since they contain two metal ions in their active sites.

The obtained results can be treated as a first step towards a better understanding of the preferences of the four mentioned enzymes towards aminophosphonate inhibitors and as a preliminary comparison of plant and mammalian enzymes.

## 2. Results and Discussion

The racemic analogues of phenylglycine studied in this work contain aromatic rings, which are variably substituted mostly with fluorine and chlorine atoms. Fluorine is considered to be a good mimic of hydrogen in medicinal chemistry, since it has been shown to be well tolerated by a variety of proteins. This results from the fact that the carbon-to-fluorine bond is only slightly (about 20%) longer than the carbon-to-hydrogen bond and thus does not introduce much steric perturbation. On the other hand, the electron-withdrawing nature of fluorine significantly affects electrostatic interactions. Chlorine is a spacious and strongly electronegative substituent and thus introduces additional steric interactions between the inhibitor molecule and the enzyme.

As seen from Table 1, nearly all of the evaluated compounds exhibited moderate micromolar inhibitory activity towards pAPN and barley enzyme and appeared to be competitive inhibitors. This is typical for low molecular-weight inhibitors, which are used as a first line to study their interactions with the enzyme. The obtained apparent inhibitory constants have to be treated with some reservation, since they cannot be attributed to either of the two enantiomers. However, modeling studies done for pAPN that are presented below have shown that, in most cases, enantiomers of the same inhibitor are bound in a very similar manner, thus indicating that their affinities should be also similar.

The comparison of the inhibitory potency against mammalian and plant aminopeptidases indicated that barley enzyme is less susceptible to the action of these aminophosphonates. This is clearly visible when comparing the effects of individual compounds. The general pattern of activity found for both enzymes seems to be quite similar. Thus, compounds **7** and **8** were the most potent inhibitors and possessed similar inhibitory potency, whereas practically no inhibitory action was observed in the case of compounds **12** and **13**. The biggest differences were obtained for compounds **3**, **4**, **10**, **11**, and **14**. These differences most likely reflect the differences in the architecture of hydrophobic pockets of the two enzymes.

The binding mode of the studied inhibitors to pAPN was further studied by molecular modeling. In the first step of the procedure, the poses docked at the distance of 20 Å from the centroid metal ion were considered. The computed induced fit docking algorithm indicated which of the amino acids were well scored, and these were therefore considered in the next step (see Appendix A).

The nature of the binding of phosphonic acid analogues of phenylglycine to pAPN was, with the exception of compounds **13** and **14**, highly uniform (Figure 1). Obviously, the dominant component of the total interaction energy was the electrostatic one between the phosphonate anion and zinc cation. Thus, the phosphonate group was involved in bidental zinc ion complexation and additionally formed hydrogen bonds with Tyr472. A strong hydrogen bond was also formed between Glu384 and the amine moiety of the inhibitor. This pattern of interactions was primary responsible for the binding of all the studied compounds.

The main feature of pAPN is its hydrophobic S1 pocket, which binds N-terminal hydrophobic amino acids of hydrolyzed peptides and proteins. Mainly, side chains of Ala348, Ala361, and Val 380 form this spacious pocket. The placement of aromatic portions of inhibitors in this binding site has a significant effect on inhibition. Thus, aromatic fragments of individual phenylglycine analogs filled this spacious pocket (with the exception of compounds **13** and **14**) in a slightly different manner, which was mostly dependent on the substitution of their phenyl ring. The examination of binding of pairs of enantiomers of individual compounds indicated that differences in their affinities were only slightly dependent on the absolute configuration of the α-carbon (as aminophosphonic acids’ *S* configuration relates to a natural *L* one). This is in agreement with reports in the literature that the inhibitory potencies of low-molecular inhibitors to aminopeptidases are only slightly dependent on their configuration, and that the expansion of their structure promotes stronger dependence [21].

The general tendency in the binding mode of the studied inhibitors is well exemplified by the comparison of the binding of the enantiomers of compounds **7** and **10** (Figure 2)—the compounds that exhibited the highest affinities towards pAPN (Table 1). Enantiomers of each of the two compounds presented only slightly different positions in the active site, with the aminophosphonate fragment of the molecules being bound in exactly the same manner. Thus, it may be safely regarded that the substitution pattern of the aromatic ring was mostly responsible for the observed differences in observed inhibitory constants. For example, the interactions of the trifluoromethyl group of compound **10** with carbonyl oxygen of Gly347 in the case of both enantiomers (Figure 2C,D), and with Arg376, as seen for isomer *S* (Figure 2D), might be responsible for the slightly reduced activity of this compound in relation to compound **7**, which was lacking these interactions.

Quite surprisingly, the calculations predicted a completely different mode of binding of the *S* enantiomers of compounds **13** and **14** (Figure 1). Their aromatic fragments were bound in the S1′ pocket of the pAPN, namely the hydrophobic pocket involved in binding the C-terminal side of the hydrolyzed peptide substrates. The difference in the architecture of the complexes of porcine aminopeptidase with enantiomers of compound **14** is shown in Figure 3. The *R* enantiomer of this inhibitor was bound canonically, similar to other studied compounds. The predicted binding of the aromatic part of isomer *S* was completely different, despite the fact that the aminophosphonate part of this molecule is normally placed in the active site. Consequently, the inhibition constants for enantiomeric forms of compounds **13** and **14** have to be significantly different.

In the MEROPS peptidase information database [22], each protease is assigned to a certain family on the basis of statistically significant similarities in their amino acid sequence, and families that are thought to be homologous are grouped together into clans. Inhibitors designed to interact with APN enzymes (belonging to the M1 family) are known to exert also an inhibitory activity versus other zinc-dependent metallopeptidases. The most commonly studied one in this respect is bovine lens leucine aminopeptidase (blLAP, M17 enzyme), which contains two zinc ions in its active site [23]. pAPN and blAP are reported to display a similar activity relationship towards synthetic substrates and inhibitors and are quite frequently mismatched [24]. In order to compare pAPN and blLAP, we also modeled the affinity of the studied compounds to blLAP.

As seen from Figure 4A, the dominant component of the total interaction energy between the inhibitor and the blLAP was also of an electrostatic nature, and results from the interaction between phosphonate dianion and two zinc cations. As shown earlier, the phosphonate anion was bound slightly more strongly to one of the zinc ions [24]. The phosphonic acid portion of the ligand was additionally stabilized by two lysines: Lys250 and Lys 262. Amino groups of phenylglycine analogs were bound either to carbonyl oxygen derived from the peptide bond of Thr259 (mostly in the case of *S*-isomers) or to Met270 (for *R*-isomers). The aromatic parts of the studied inhibitors selected the hydrophobic S1 pocket, which was built mainly by side chains of two alanines and two methionines. Additional interactions with other amino acids enhanced the binding of individual compounds (Figure 4A).

The bindings of the two inhibitors ranking as the best for both pAPN and barley enzyme (compounds **7** and **8**) are presented in Figure 5. The best poses showed that their phosphonic moieties interacted with zinc ions via strong electrostatic interactions. The observed distances of phosphonate anions of both enantiomers of compound **7** were found to be identical and equal for Zn488 (2.06 Å) and for Zn489 (2.13 Å). Aromatic fragments of inhibitors bound canonically with Ala451, playing a key role in binding *R*-isomers. Thus, in the case of blLAP, the predicted binding of the enantiomeric forms of inhibitors differed to a great extent in comparison with pAPN.

Genome analysis has indicated that plants, like animals, possess a variety of aminopeptidase genes; however, they have not been characterized suitably at the enzyme level to date [25]. Since the three-dimensional structures of plant enzymes, with the exception of tomato wound-induced enzyme [20], are not known, it is only postulated that most plant enzymes belong to the class of leucine aminopeptidases (LAPs). Thus, in plants, there are basically two classes of LAPs [26,27]: the neutral ones (APN and its orthologs), which are constitutively expressed and detected in all plants, and the stress-induced acidic ones (LAPA), which are expressed only in a subset of the *Solanaceae*.

The crystal structure of the leucine aminopeptidase from barley is not yet known. To compare molecular modeling studies with experimental data, the studied inhibitors were also docked to the active center of tomato leucine aminopeptidase (tLAPA), the only plant aminopeptidase with a known crystal structure. We supposed that tomato and bovine lens enzymes are the closest structures described so far. Both of them contain two metal ions in the catalytic center and both belong to the M17 family. The mammalian aminopeptidase is a zinc-dependent enzyme, whereas the plant one is magnesium-dependent.

The comparison of the binding mode of phosphonic acid analogs of phenylglycine with blLAP and tLAPA (Figure 4) indicates that it was almost the same for the two enzymes. Metal ions were held by aspartic and glutamic acids. The phosphonate anions of the ligands were additionally stabilized by two lysines, while the amine interacted with threonines or methionines, as described earlier for blLAP.

Deeper insight into the mode of the predicted tLAPA–inhibitor architecture provided the analysis of the binding of compounds **7** and **8**, depicted in Figure 6. Also, in this case, the phosphonate anion was bound slightly more strongly to one of the magnesium ions, with the distances between the enantiomers of compound **7** being nearly the same, namely 2.06 Å and 1.98 Å (for *R* inhibitors) and 2.06 Å and 1.95 Å (for *S* inhibitors).

The hydrophobic pocket of the enzyme was also composed of two alanines and two methionines; however, some minute differences between the modes of binding were also visible. These were the presence of methionine 460 interacting with the phenyl ring of the *S*-isomer of the two inhibitors, and the interaction of the guanidine group from Arg360 with the chlorine atom of *R*-isomers. This indicates that the binding of individual compounds is highly unified if compared with pAPN.

## 3. Conclusions

It is believed that implementation of computer-aided drug design techniques could help complement the experimental methods and facilitate ligand/structure-based design. Low-molecular enzyme inhibitors are commonly considered as useful for deepening our understanding of the enzyme architecture and thus may serve as a first step in designing more complex, more effective, and specific inhibitors with enhanced activity. The lower specificity of the inhibitors towards the enzymes was reflected by the inhibition constant in the micromolar range. The studies presented in this work, although in good agreement with the general scheme of inhibitor design, show that this concept might not be fully applicable. Our results indicated the different mode of interaction of single phosphonic acid analogues of phenylglycine with the enzyme spacious hydrophobic site, which binded aromatic portions of these molecules. Such a dispersion of binding modes of relatively simple and structurally similar compounds caused purely modeling-based lead optimization to be difficult for these compounds, requiring either additional experimental proof to support the binding poses or the collection of crystal structures of enzyme-inhibitor complexes to provide additional insight into their architecture.

Also, the use of the knowledge of the structures of related enzymes (in our case, two mammalian and plant enzymes) and their complexes with a certain set of inhibitors did not allow even for speculative reasoning regarding the structure of enzymes with an unknown crystal structure (in this case, barley aminopeptidase).

## 4. Materials and Methods

### 4.1. Compounds

Aminobenzylphosphonic acids were available from previous studies [13] and were synthesized based on the procedure described by Oleksyszyn and Soroka [28,29,30]. All compounds used in this work were racemic mixtures in the form of crystalline solids and could safely be stored at room temperature for several months.

### 4.2. Enzyme Preparations

Microsomal leucine aminopeptidase (APN, EC 3.4.11.2) from porcine kidney was purchased from Sigma Aldrich. It was applied directly in kinetic measurements after dissolving in 50 mM potassium phosphate (pH 7.2). Such a solution should be stored in 4 °C for no longer than 2 days.

Aminopeptidase from barley seeds (*Hordeum vulgare* L.) was isolated and purified based on the procedure described previously [18]. The following steps were used: (i) ammonium sulfate precipitation, (ii) gel chromatography (Sephadex G-25, Sephacryl HR 300), followed by (iii) ion chromatography (DEAE—Sepharose). The molecular mass of the extracted enzyme was ~58 kDa. Because of the low stability of the concentrated enzyme solution, it was prepared directly before each kinetic study.

### 4.3. Kinetic Characterization of Aminopeptidases

The activity of pAPN (K_M_ = 0.52 mM) was determined as described in the literature [31]. The activity of aminopeptidase from barley seeds was determined at 37 °C in 50 mM Tris-HCl, pH 8.0, containing 50 mM NaCl and 10 mM β-mercaptoethanol using substrate *L*-leucine-*p*-nitroanilidine (*L*-Leu-pNa) dissolved in DMSO. The kinetics parameters of the purified enzyme were measured for *L*-Leu-pNa at ten final concentrations (ranging from 0.1 to 1.0 mM) and being repeated two times. The K_M_ value of barley seeds aminopeptidase was determined by using the Lineweaver–Burk weighted regression method (see Appendix A).

### 4.4. Inhibitory Studies

The assay mixture of the pAPN, totaling 1.01 mL, contained 50 mM potassium phosphate buffer (pH 7.2), 0.025 mL of the substrate *L*-Leu-pNa dissolved in DMSO (used in 0.4–0.1 mM range of the final concentration), 0.05 mL of the potential inhibitor in reaction buffer (in a concentration dependent on the potency of the compound), and 0.01 mL of the enzyme solution (4µg/mL final concentration).

The assay mixture of the aminopeptidase from barley seeds, totaling 1.095 mL, contained 50 mM Tris-HCl buffer (pH 8.0) with 50 mM NaCl and 10 mM β-mercaptoethanol, 0.025 mL of the substrate *L*-Leu-pNa dissolved in DMSO (used in 0.4–0.1 mM range of the final concentration), 0.05 mL of varying concentrations of the potential inhibitor (dependent on the potency of the compound), and 0.02 mL enzyme (0.030 mg of protein).

The progress of the enzymatic reaction for both enzymes was measured at 37 °C for 10 min and was monitored spectrophotometrically (UV-VIS Spectrophotometer, JASCO V-730) by following the change in absorbance at 405 nm (formation of *p*-nitroaniline). The extinction coefficient for *p*Na was 9620 M^−1^cm^−1^.

Inhibition constants were determined considering four substrate concentrations and five concentrations of each inhibitor, with each measurement being repeated at least twice.

The rate of enzymatic reaction was studied following the reaction progress (changes in absorbance of *L*-Leu-pNa over time). Kinetics constants, namely K_i_ and IC_50_, and the type of the inhibition were determined using Lineweaver–Burk, Dixon, and Hanes–Wolf procedures. For each parameter, relative errors were calculated. In the case of K_i_ and IC_50_, the range of errors did not exceed 10% for each of the tested inhibitors.

### 4.5. Molecular Modeling

Crystal structures of enzymes were obtained from the Research Collaboratory for Structural Bioinformatics Protein Data Bank (RCSB-PDB); bovine (*Bos taurus*) M17 aminopeptidase 1LAM [32], porcine (*Sus scrofa*) M1 aminopeptidase 4FKE [33], and tomato (*Solanum lycopersicum*) acidic leucine aminopeptide 4KSI [20]. The water and ligands were removed and the proteins were protonated in experimental pH [34]. Before docking, the structures of the inhibitors and their stereochemistry were considered, protonated in experimental pH typical for each enzyme also, and optimized by LigPrep [35]. All of the compounds were docked with the use of the induced fit docking [36] algorithm of the Maestro Schrodinger Package. The VSGb (variable-dielectric generalized Born) model was used, which incorporates residue-dependent effects. The solvent was water. Centroid positions of metal ion(s) with the range of 20.0 Å were selected as the docking box center. Ligands were docked with the sample ring conformations option with a 2.5 kcal/mol energy window and standard glide, prime refinement and glide redocking (SP) procedures for the best pose for each compound. MM-GBSA (molecular mechanics-generalized born surface area) was performed as rePrime refinement [36] to calculate Gibbs free energies with protein flexibility, with the distance from ligands also set as 0.0 Å and 5.0 Å. The first pose with the lowest binding energy in 5.0 Å was selected as the best one.

## Figures and Tables

**Figure 1 pharmaceuticals-12-00139-f001:**
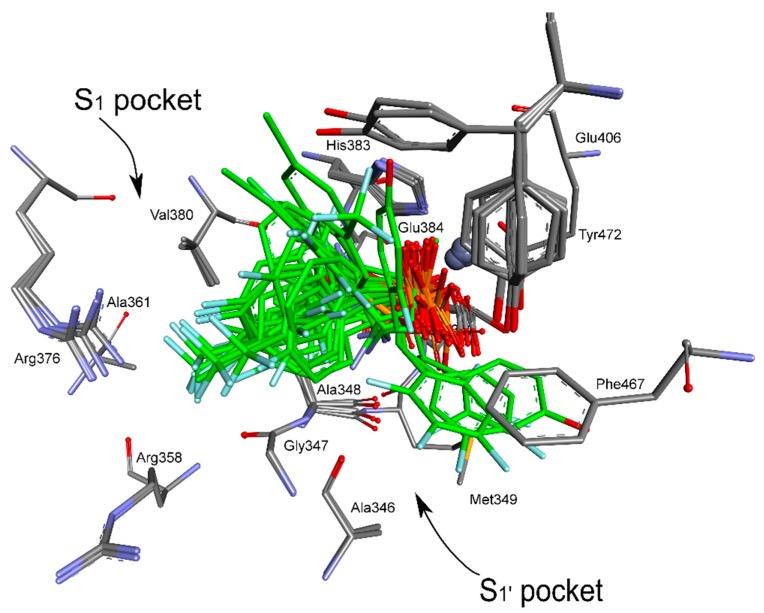
Modes of binding of all the studied analogs of phenylglycine (both enantiomers shown for each compound) by porcine aminopeptidase (pAPN). The carbon and chlorine atoms of inhibitors are colored green, phosphonate oxygen atoms in red, and the amine group in blue. Fluorine is indicated as light blue. The zinc ion is shown as a grey sphere.

**Figure 2 pharmaceuticals-12-00139-f002:**
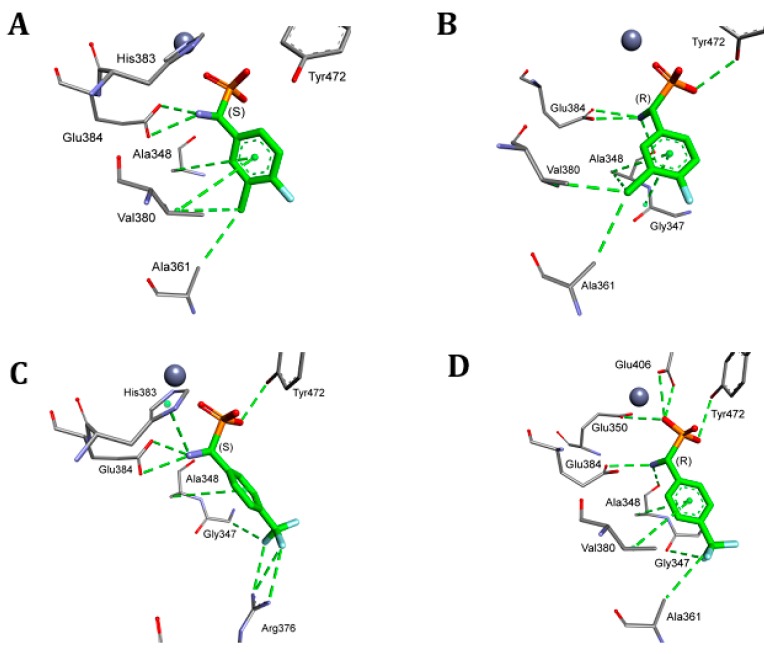
Mode of binding of both enantiomers of representative examples of phosphonic analogues of phenylglycine to porcine alanyl aminopeptidase (pAPN). **A**: *S*-isomer, **B**: *R*-isomer of compound **7**, **C**: *S*-isomer, **D**: *R*-isomer of compound **10**. All of the amino acids interacting even with even one of the inhibitors are shown. Atoms are colored as in Figure 1.

**Figure 3 pharmaceuticals-12-00139-f003:**
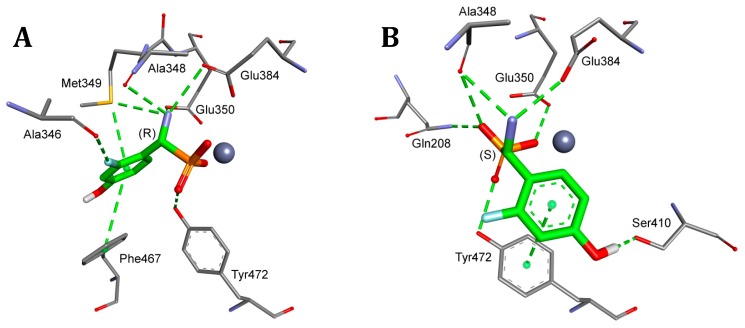
Mode of binding of the two enantiomers of compound **14** by porcine aminopeptidase (pAPN): **A**—*R* isomer, **B**—*S* isomer. The oxygen atom of hydroxyl is shown in red.

**Figure 4 pharmaceuticals-12-00139-f004:**
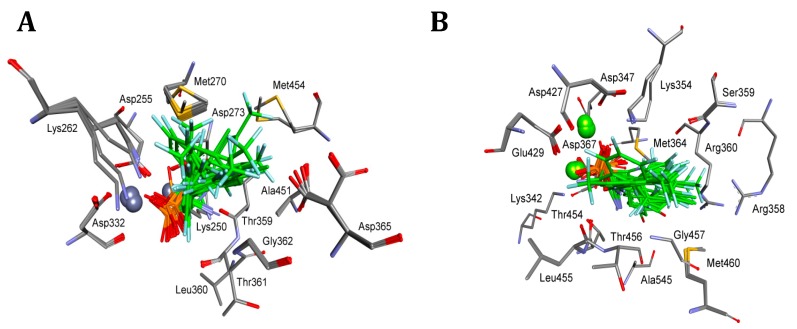
Modes of binding of all the studied analogs of phenylglycine (both enantiomers shown for each compound) to **A**—bovine lens leucine aminopeptidase (blLAP) and **B**—tomato leucine aminopeptidase (tLAP). All of the amino acids interacting with even one of the inhibitors are shown. Magnesium ions are shown as green spheres; other atoms are colored as in Figure 1.

**Figure 5 pharmaceuticals-12-00139-f005:**
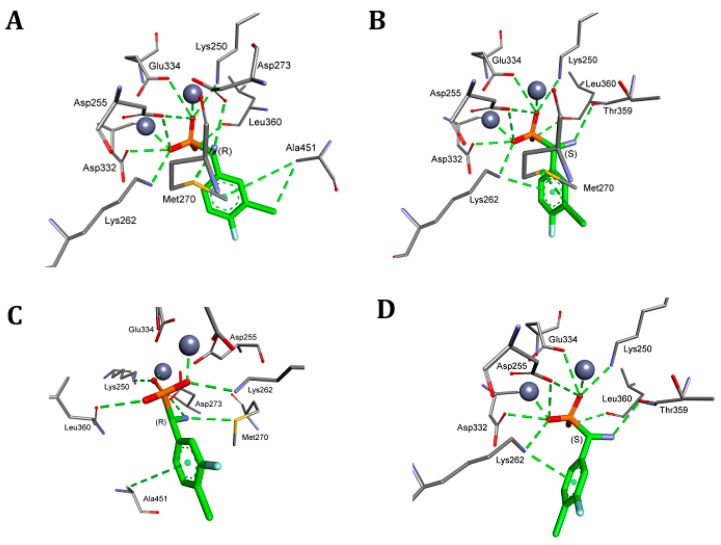
Mode of binding of both enantiomers of representative analogues of phenylglycine by bovine lens leucine aminopeptidase. **A**: *S*-isomer, **B**: *R*-isomer of compound **7**, **C**: *S*-isomer **D**: *R*-isomer of compound **8**.

**Figure 6 pharmaceuticals-12-00139-f006:**
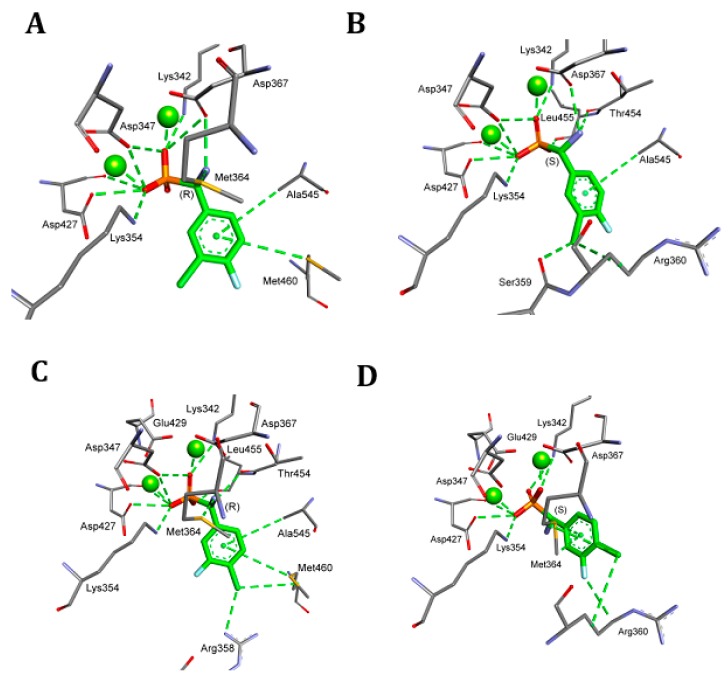
Mode of binding of both enantiomers of compounds **7** and **8** by tomato leucine aminopeptidase. **A**: *S*-isomer, **B**: *R*-isomer of compound **7**, **C**: *S*-isomer, **D**: *R*-isomer of compound **8**.

**Table 1 pharmaceuticals-12-00139-t001:** Inhibitory potencies of phosphonic acid analogues of phenylglycine towards porcine aminopeptidase (pAPN) and aminopeptidase from barley seeds.

Com-Pound	Structure	pAPN K_i_ [μM] + SD ^1^	Barley Seed AP K_i_ [μM] + SD ^1^	Com-Pound	Structure	pAPN K_i_ [μM] + SD ^1^	Barley Seed AP K_i_ [μM] + SD ^1^
**1**	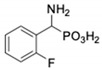	232 ± 22	69 1 ± 68	**8**	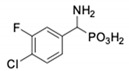	69.1 ± 6.5	103 ± 10
**2**	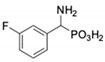	122 ± 10	311 ± 27	**9**	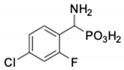	299 ± 11	818 ± 43
**3**	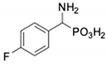	123 ± 2.7	1105 ± 57	**10**	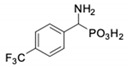	56.3 ± 2.2	686 ± 70
**4**	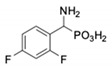	258 ± 23	1425 ± 83	**11**	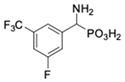	164 ± 7.7	1033 ± 73
**5**	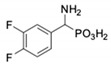	101 ± 3.8	549 ± 54	**12**	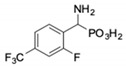	in ^2^	in ^2^
**6**	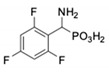	in ^2^	in ^2^	**13**	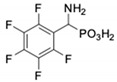	in ^2^	in ^2^
**7**	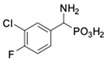	45.4 ± 4.5	62.1 ± 2.3	**14**	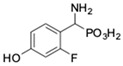	116 ± 22	in ^2^

^1^ standard deviation (SD) of the K_i_ values. ^2^ inactive at a concentration of 1 mM. ^3^ IC_50_ = 2.43 mM. ^3^ IC_50_ = 1.03 mM.

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
