# Peer review of "Phosphonic Acid Analogues of Phenylglycine as Inhibitors of Aminopeptidases: Comparison of Porcine Aminopeptidase N, Bovine Leucine Aminopeptidase, Tomato Acidic Leucine Aminopeptidase and Aminopeptidase from Barley Seeds"

_pharmaceuticals, 2019, doi:10.3390/ph12030139_

Round 1

Reviewer 1 Report

The manuscript requires a significant rewrite before it can be effectively assessed. 

Author Response

The manuscript requires a significant rewrite before it can be effectively assessed.

The manuscript was significantly changed and we do hope that it is suitably improved now.

Reviewer 2 Report

In this work, the authors assess the inhibition activity of a set of racemic phosphonate analogues of phenylglycine towards porcine and barley aminopeptidases. They also perform in silico modeling on each enantiomer to determine the preferred binding on porcine, barley, bovine and tomato aminopeptidases. Given the limited insight provided by a purely in silico docking procedure, the work has a limited utility due to the lack of experimental structural data (e.g. by X-ray crystallography or NMR spectroscopy). Furthermore, binding affinities cannot be considered reliable due to the fact that racemic mixtures of each compound were employed. However, it can be considered as a first step towards a better understanding of the substrate preferences of these enzymes.  

The manuscript feels unpolished and hard to read, and should be revised by improving the flow of the concepts in the introduction and in the abstract, and by checking the language and grammar. 

A few specific issues should be addressed before the manuscript can be considered for publication. 

Major issue: 

1) It is mentioned (only once!) that the compounds tested are racemic mixtures. As a consequence of this, the Ki values obtained in vitro are only apparent and cannot be attributed to either of the two enantiomers. Moreover, they hide any information concerning the actual enantiomer selectivity (i.e. how different are the Ki for S and R enantiomers). Therefore, all the results obtained should be discussed more carefully in this respect. 

2) The authors find that the major energy contribution comes from electrostatics. Are these calculated in vacuum or in implicit or explicit solvent? In void, it is rather obvious that the electrostatic interactions would be dominant. This should be clarified in the methods section. 

Minor comments: 

1) the language and phrasing are sometimes unclear throughout the manuscript, making it sometimes hard to understandA few examples: 
At line 17, “Although both enzymes differ” should be “Although the two enzymes differ” 
At line 33, “low-molecular enzyme inhibitors” maybe the authors meant “low-molecular weight enzyme inhibitors” 
At line 40: “towards series of their substrates” The meaning should be clarified, were the activities compared for the same substrates, or for different ones? 
At line 73: “In order to understand better the binding mode of the studied inhibitors it was further studied by molecular modeling using pAPN” the phrase is not clear. 
Many other typos missing characters and misplaced punctuation are present all over the text. 

2) The abstract is quite confusing, and the overall scope of the work is not conveyed clearly. I suggest that it is rewritten following more closely what is stated in the conclusions section, which is clearer.  
Also, it is not clear in the abstract and in the introduction for which enzymes in vitro activities were measured and for which enzymes molecular modeling was carried out. These parts should be rewritten to clarify this.  

3) In the abstract: "Modeling of their binding to the acidic leucine amino peptidase from tomato did not allow speculating if the barley enzyme is mono or bi-zinc one.” I could not find in the main text any reference to this kind of speculation about metal coordination. That should either be discussed in the main text or removed from the abstract. 

4) The name of the software used for the modeling should be removed from the abstract, as it may sound like an advertisement of a commercial software. It is correct and sufficient to mention it in the methods section. 

5) Naming consistency should be improved: sometimes the enzymes are called “amino peptidase”, sometimes “aminopeptidase”, sometimes “metalloaminopeptidase”. Also, only acronyms should be used throughout the text after they have been defined the first time (see for example “porcine aminopeptidase” at line 79). 

6) At line 38: “aminopeptidase N (alanine aminopeptidase, pAPN) from porcine kidney [11,14,15], which is a good model enzyme for human aminopeptidase” It should be specified for which isoform(s) of human aminopeptidase pAPN is considered a good model.  

7) At line 71 “The biggest differences were obtained for compounds 3, 4, 10 and 14”: also compound 11 exhibits quite a large difference.  

8) At line 75 “, an enzyme bearing one zinc ion in the active site” that is already stated at line 44.  

9) At line 77, the acronym “IFD” is not defined. 

10) In Table 1, when not specified otherwise, what is the maximum concentration at which unactive (inactive?) compounds have been tested? 

11) Figure 4B is called in the text after Figure 5, therefore their order should be changed accordingly. 

Author Response

We would like to thank the Reviewer for useful and thoughful comments. Here are the responses:

Major issues: 

1) It is mentioned (only once!) that the compounds tested are racemic mixtures. As a consequence of this, the Ki values obtained in vitro are only apparent and cannot be attributed to either of the two enantiomers. Moreover, they hide any information concerning the actual enantiomer selectivity (i.e. how different are the Ki for S and R enantiomers). Therefore, all the results obtained should be discussed more carefully in this respect. 

We have changed the layout of the paper taking into consideration stereochemical issues and discussed the meaning of apparent inhibitory constants in some detail in both - part devoted to experimental as well as modeling description.

2) The authors find that the major energy contribution comes from electrostatics. Are these calculated in vacuum or in implicit or explicit solvent? In void, it is rather obvious that the electrostatic interactions would be dominant. This should be clarified in the methods section. 

Calculations have been done in water - this information was inserted into Experimental. We were that it is obvious that electrostatic interactions have to be dominant and expressed it in the manuscript.

Minor comments: 

1) the language and phrasing are sometimes unclear throughout the manuscript, making it sometimes hard to understandA few examples: 
At line 17, “Although both enzymes differ” should be “Although the two enzymes differ”; At line 33, “low-molecular enzyme inhibitors” maybe the authors meant “low-molecular weight enzyme inhibitors” ;At line 40: “towards series of their substrates” The meaning should be clarified, were the activities compared for the same substrates, or for different ones? ; At line 73: “In order to understand better the binding mode of the studied inhibitors it was further studied by molecular modeling using pAPN” the phrase is not clear. 
Many other typos missing characters and misplaced punctuation are present all over the text. 

We have carefully examined manuscript in order to improve English and to eliminate typoghraphic errors. I hope that with success. Of course we have applied to all the comments of the Reviewer 2.

2) The abstract is quite confusing, and the overall scope of the work is not conveyed clearly. I suggest that it is rewritten following more closely what is stated in the conclusions section, which is clearer.  
Also, it is not clear in the abstract and in the introduction for which enzymes in vitro activities were measured and for which enzymes molecular modeling was carried out. These parts should be rewritten to clarify this.  

We have completely rewritten the abstract taking into consideration of this comment

3) In the abstract: "Modeling of their binding to the acidic leucine amino peptidase from tomato did not allow speculating if the barley enzyme is mono or bi-zinc one.” I could not find in the main text any reference to this kind of speculation about metal coordination. That should either be discussed in the main text or removed from the abstract. 

and 4) The name of the software used for the modeling should be removed from the abstract, as it may sound like an advertisement of a commercial software. It is correct and sufficient to mention it in the methods section. 

These fragments were removed from the abstract

5) Naming consistency should be improved: sometimes the enzymes are called “amino peptidase”, sometimes “aminopeptidase”, sometimes “metalloaminopeptidase”. Also, only acronyms should be used throughout the text after they have been defined the first time (see for example “porcine aminopeptidase” at line 79). 

I hope that we have conformed to this comment

6) At line 38: “aminopeptidase N (alanine aminopeptidase, pAPN) from porcine kidney [11,14,15], which is a good model enzyme for human aminopeptidase” It should be specified for which isoform(s) of human aminopeptidase pAPN is considered a good model.  

We have specified in the text that this refers to human alanine aminopeptidase

7) At line 71 “The biggest differences were obtained for compounds 3, 4, 10 and 14”: also compound 11 exhibits quite a large difference.  

8) At line 75 “, an enzyme bearing one zinc ion in the active site” that is already stated at line 44.  

9) At line 77, the acronym “IFD” is not defined. 

We have conformed to these comments

10) In Table 1, when not specified otherwise, what is the maximum concentration at which unactive (inactive?) compounds have been tested? 

It is already given under the Table as 1 mM

11) Figure 4B is called in the text after Figure 5, therefore their order should be changed accordingly. 

The numbering of Figures has changed upon changed layout of the manuscript

Reviewer 3 Report

The authors present molecular docking study on several phosphonic analogues of phenylglycine as potent inhibitors of porcine aminopeptidase N.  The results were compared with the results on other aminopeptidases. I have found the study quite interesting, however, I have some questions and comments.

- It is clearly visible that the binding modes (binding interactions) of both enantiomers of all studied phenylglycines to various substrates are different from each other, which means that their affinities towards the substrates must be different. This is a little bit in contradiction with the statement in the Introduction that the binding affinities of both enantiomers is similar. I think that this should be somehow explained.

- I do not understand why the experimental section contains the description of experimental procedures that are not mentioned in the Results. They have no relation to the main topic of the manuscript and, according to my opinion, should not be there.

- I appreciate very critical evaluation of the results that was expressed in the Conclusions. However, as it is formulated in the manuscript, it looks like that the authors have some doubts, whether the study is worth of publishing, which I don’t share. I do think that the manuscript is worth of publishing.

The manuscript is written clearly and with a minimum of typographical errors. The English is very decent and does not need corrections. Nevertheless, I have found some typos, which are listed below, together with other comments:     

80: 1nd - and 131: Fig. 2 is a little bit cluttered to be able to demonstrate a different binding mode of compounds 13 and 14. I would suggest to simplify it by showing, e.g. compounds 7 and 10 as examples of let’s say a typical binding mode, and 13 and 14 as the extraordinary one 159; the same is true for Fig. 4. Although the binding mode is visible, I would prefer certain simplifying 167: bots –both 167: …to be identical and equal…. – to be almost identical…. 169: in the binding of R isomers 212: The sentence “Being only roughly specific towards the enzyme they usually exhibit the effect at micromolar range.“ is not very clear to me. I suggest to rewrite it, e.g. in this way: Lower specificity of the inhibitors towards the enzymes is reflected by the inhibition constant in the micromolar range. 214: might be not very useful - might not be very useful 233: the verb might should be replaced with should 234: basing – based 249: change - the change

Author Response

1./ We would like to tha nk the reviewer fo valuable and nice comments, which enabled to improve the paper. Below are answers to these comments:It is clearly visible that the binding modes (binding interactions) of both enantiomers of all studied phenylglycines to various substrates are different from each other, which means that their affinities towards the substrates must be different. This is a little bit in contradiction with the statement in the Introduction that the binding affinities of both enantiomers is similar. I think that this should be somehow explained.

We have changed the layout of the paper taking into consideration interaction between enzymes and enantiomers of phenylglycine analogs. These issues are now more detaily disscussed

2./ I do not understand why the experimental section contains the description of experimental procedures that are not mentioned in the Results. They have no relation to the main topic of the manuscript and, according to my opinion, should not be there.

We have removed these data from Experimental

3./ I appreciate very critical evaluation of the results that was expressed in the Conclusions. However, as it is formulated in the manuscript, it looks like that the authors have some doubts, whether the study is worth of publishing, which I don’t share. I do think that the manuscript is worth of publishing.

We would like to thank Reviewer 3 for this well-wishing comment.

4./ The manuscript is written clearly and with a minimum of typographical errors. The English is very decent and does not need corrections. Nevertheless, I have found some typos, which are listed below, together with other comments: 80: 1nd - and 131: Fig. 2 is a little bit cluttered to be able to demonstrate a different binding mode of compounds 13 and 14. I would suggest to simplify it by showing, e.g. compounds 7 and 10 as examples of let’s say a typical binding mode, and 13 and 14 as the extraordinary one 159; the same is true for Fig. 4. Although the binding mode is visible, I would prefer certain simplifying 167: bots –both 167: …to be identical and equal…. – to be almost identical…. 169: in the binding of R isomers 212: The sentence “Being only roughly specific towards the enzyme they usually exhibit the effect at micromolar range.“ is not very clear to me. I suggest to rewrite it, e.g. in this way: Lower specificity of the inhibitors towards the enzymes is reflected by the inhibition constant in the micromolar range. 214: might be not very useful - might not be very useful 233: the verb might should be replaced with should 234: basing – based 249: change - the change

We have tried to improve English of the manuscript taking into consideration all the comments of the Reviewer. All indicated errors have been corrected.

If considers Figures 2 and 4 (now 1 and 4) we have tried to show that most of the inhibitors are bound canonically and rather standardly. Figure 1 also nicely indicates different binding of S isomers of compounds 13 and 14. Figure 4 was designed to indicate thesimilarity of overall binding modes between blLAP and tALAP. We do feel that they well serve this purposes.

Round 2

Reviewer 1 Report

The manuscript by Wanat et al describes the comparison of phosphonic acid analogues of phenylglycine as inhibitors of aminopeptidases from bovine, tomato, and barley seed. I recommend for publication after minor revisions

General: Throughout (including title) have "phosphonic analogues..." should be changed to "phosphonic acid analogues...."

Results and Discussion

Table 1: is the error reported in the table for the Ki calculation standard deviation? This should be noted as a footnote at bottom of table.

Page 3: have " ...has a significant effect on an inhibition......" should be "...has a significant effect on inhibtion."

Figures 2,3,4,5,6: The amino acid residue labels are very tiny and hard to read. Font size should be increased. 

Page 6, paragraph 2: "As seen from Figure 4A, also in this case is....." This sentence is awkward please rewrite. 

Page 8, paragraph 1: Last sentence is awkwardly written and should be rephrased.

Materials and Methods

4.1 Compounds. 

How the compounds were stored would be worthwhile information to include for the reader.

4.2 Enzymatic preparations

While the purification has been reported elsewhere the authors should briefly describe the preparation.

4.3 Kinetic characterization

Km values do not belong in the methods section. This should be put in a table in the supplementary material.

The assay should be briefly described (see above)

How many replicates were performed (biological and technical)? this should be stated. Regression results should be provided in supplementary material.

4.4 Inhibitory studies

Subtstrate concentrations presented are these final concentrations? This should be clarified. What type of error was calculated (standard error of mean, standard deviation?) The error value is meaningless unless the number of biological and technical replicates is stated.

Supplemental Information

Table titles need to be more descriptive. 

Author Response

General: Throughout (including title) have "phosphonic analogues..." should be changed to "phosphonic acid analogues...."

done

Results and Discussion

Table 1: is the error reported in the table for the Ki calculation standard deviation? This should be noted as a footnote at bottom of table.

Table was supplemented by these data

Page 3: have " ...has a significant effect on an inhibition......" should be "...has a significant effect on inhibtion."

correctede

Figures 2,3,4,5,6: The amino acid residue labels are very tiny and hard to read. Font size should be increased. 

We have done it

Page 6, paragraph 2: "As seen from Figure 4A, also in this case is....." This sentence is awkward please rewrite. 

Sentence was rewritten

Page 8, paragraph 1: Last sentence is awkwardly written and should be rephrased.

Sentence was rephrased

Materials and Methods

4.1 Compounds

How the compounds were stored would be worthwhile information to include for the reader.

This information was provided

4.2 Enzymatic preparations

While the purification has been reported elsewhere the authors should briefly describe the preparation.

Suitable information was added. 

4.3 Kinetic characterization

Km values do not belong in the methods section. This should be put in a table in the supplementary material.

These data alongside with suitable graphs are added to supplementary material

The assay should be briefly described (see above)

These data are introduced into the text

How many replicates were performed (biological and technical)? this should be stated. Regression results should be provided in supplementary material.

The representative data are shown in supplementary material

4.4 Inhibitory studies

Subtstrate concentrations presented are these final concentrations? This should be clarified. What type of error was calculated (standard error of mean, standard deviation?) The error value is meaningless unless the number of biological and technical replicates is stated.

The description of kinetic studies was significantly rewritten and reference 31 exchanged for one yielding moire detailed procedurę.

Supplemental Information

Table titles need to be more descriptive. 

We have changed these titles

This manuscript is a resubmission of an earlier submission. The following is a list of the peer review reports and author responses from that submission.

Round 1

Reviewer 1 Report

Review of the paper “Phosphonic analogues of phenylglycine as inhibitors of aminopeptidases: Comparison of porcine aminopeptidase N, bovine leucine aminopeptidase  and aminopeptidase from barley seeds”

I find this paper interesting and without doubt in focus of the journals scope. Below I provide several comments that should be addressed before the paper is ready for publication.

I have split my comments into the review below as well as remarks in the pdf of the manuscript in a form of yellow-marked comments.

Results

The authors provide linear correlation between DG of binding of ligand and experimentally determined Ki. A separate correlations are provided for R and S enantiomers.

I have some methodological problems with this analysis. From the table 1 one can suspect that experimental values were collected for racemic mixture of the compounds not the pure enantiomers. This issue is not clear from reading the experimental section. Please indicate what was the experimental protocol.

If the Ki were determined by racemic mixtures of inhibitors the obtained Ki value may be a superposition of interactions of both enantiomers or just one of them if the KiS/KiR were far from 1.

The authors should appropriately comment this in the manuscript text.

In the section that compares binding mode of S- and R-7 and 10 the hypothetical reasons for differences in DG of binding must be supported by quantitative interaction analysis which could support authors thesis. It would be best if such partitioning of interaction energy was provided for vdW and electrostatic interactions between important residues and Zn2+ ion and ligand. Later in the text such analysis is mentioned so the authors indeed have these data.

Molecular modeling

I have several question concerning modeling protocol.

1.       Please provide data on the selected protonation state of the protein (charge, number of atoms, preferentially the protein with some exemplary docking pose in the supplement)

2.       How many poses were obtained and characterized with DG binding?

3.       What is the variability of the DG observed for the different poses. Normally MM-GBSA or MM-PBSA is conducted for stable (in terms of RMSD) production phase of MD simulation and as a result DG is obtained with RMS error resulting from thermal motion of both protein and ligand (and different G for different conformations). Here only singular values are provided, without any proof that good sampling was achieved and no information on distribution of DG biding. Perhaps differences between enantiomers’ DGs  are well within error of the estimation method? As the ‘dispersion of binding modes’ is a part of the conclusions of the paper this issue should be addressed with a bit more care.

4.       What portion of the enzyme was flexible in the Induced Docking procedure. This procedure differs between different programs – so please clarify how it was done in the protocol (soft potentials and then optimization of flexible residues and ligands? Or perhaps conformational analysis of the receptor prior to docking and docking to different conformations of the active site?) At the current stage the procedure is not reproducible by other researchers.

5.       Has any optimization protocol (such as ligand annealing or short MD simulation) been used to ensure that poses selected by scoring function indeed were the best one (i.e. conformational sampling vs just geometry minimization down the local gradient)?

6.       Please provide results of  the ligand-protein interaction analysis which is mentioned in the text in all places suggested in the manuscripts.

Minor things:

-          Please correct the language of the paper – it is very uneven (parts are very good, parts are very awkward)

-          Please ensure that figures stay above figure’s captions and are not shifted to other places as in the provided manuscript.

- Please work on the quality of correlation plots the one used now is not very nice (e.g. Ki floating off the y-axis, R2 overlapping points and numbers, etc.) - looks like made in Excel

Reviewer 2 Report

Summary
The authors measure the inhibition of porcine aminopeptidase N and barley metalloaminopeptidase by fluorine- and chlorine-substituted phosphonic analogues of phenylglycine. The authors use docking to propose that the binding modes of their series to pAPN are similar to the binding modes of their series to bovine lens leucine aminopeptidase. The authors also obtain good correlation between experiment and relative free energy calculations performed using Schrodinger’s MM-GBSA. On the basis of the modeled binding poses, the authors provide good rationales for the trends observed in their experimental results. The authors demonstrate that even for these simple ligands, there is quite a bit of variability in the binding modes even though the differences in activity are small. However, the charts in this manuscript could use some cleaning up, and the primary conclusion needs some additional proof. Please see my comments and questions below.

Comments/Suggestions

The MM-GBSA plot needs to be redrawn. The R2 values are obscuring data points and the Y axis. The Y axis needs to be moved to the left-hand-side of the plot. The two lines seem to have shadows -- these need to be removed.

Why was MM-GBSA only calculated for 5 compounds (cmpds 7-11)? Calculations should be provided for the other compounds as well. Even in the cases of compounds 6, 12, and 13, it would be interesting use the MM-GBSA calculations and the line of best fit to predict the Ki values and see if the relative Ki values for these 3 compounds make sense.

The authors claim to have modeled the affinity of their ligands against biLAP (Page 5, line 141). The authors should present these results as a figure (preferably) or table somewhere in the manuscript.

The primary conclusion of the manuscript seems to suggest that further lead optimization based on these compounds is tough because there are big changes in binding modes for small changes in activity. A potential fallacy in this conclusion lies in assuming that the modeled binding modes are the real binding modes. I like modeling, but without co-crystal structures to provide additional proof, it would not be prudent to assume that modeled binding poses are correct! The authors should either mention this caveat in the conclusions and say that purely-modeling based lead optimization is hard for these compounds, or provide additional experimental proof to support the binding poses.

Questions
1.    Can the authors comment on potential binding modes to human aminopeptidase?
2.    In the "Molecular Modeling" subsection of the  Materials & Methods section, the authors should provide Glide and Induced Fit Docking parameters used (even if they were defaults).

Reviewer 3 Report

This manuscript might talk about interesting topics about inhibitory effect of the phosphoric analogues. However the structure and language are not well originated, the readers may hardly get any clear idea. I suggest authors carefully re-write the manuscript and submit again.

Reviewer 4 Report

This manuscript describes the evaluation of 14 phophonic analogues of phenylglycine as inhibitors of plant and animal derived aminopeptidases. Initially, authors determined the inhibition constants of 14 phosphonic analogues of phenylglycine towards pAPN and barley AP and, subsequently, analyzed the binding affinities of some molecules for the pAPN by using molecular modeling. The authors also compared the binding mode of the phosphonic analogues and their enantiomers in the active site of animal (pAPN, blLAP) as well as plant (biLAP) derived aminopeptidase.

This study showed the important step for designing inhibitors of animal and plant derived aminopeptidase.  However, many results and statements seem not to be supported by the sufficient evidences, thus, it should be further refined.

 Major points:

1.     The English in the present manuscript is not of publication quality and require major improvement.

2.     Table 1, the authors showed no experimental results of Ki values for R isomers, which seem to be used in Y axis in Fig. 1 and 6.

3.     Line 70, the authors didn’t mention why compounds 7, 8, 9, 10 and 11 were selected for molecular modeling experiments but not all molecules.

4.     Line 100, It is not clear to see “completely different binding…” in Fig. 3. Better Fig. is necessary to support this statement.

5.     Line 107, It is important to show data for differences of affinity between S- and R- isomers.

6.     Line 142, please indicate which Fig. is associated with this statement. Again, please explain why only 7, 8 9 10, 11 were chosen for the modelling otherwise all molecules should be included in the modeling experiment.

7.     Line 145, It is very important to mention what is different between the two enzymes (pAPN and blLAP).

8.     Fig. 4, there are no statements why 11 and 14 were chosen for this modeling experiments. Otherwise data for other molecules should be shown (i.e. in supplementary data)

9.     Line 172, please explain more about “significantly different manner”

10.  Fig. 5, It is difficult see the binding state of 14s. The presentation of Fig 5. needs to be modified.

11.  Line 204, 207, these two statements are contradictory, “general trend for both enzymes is quite similar” v.s. “The biggest differences were ….”. It seems that the biggest differences are observed also at 11 and 14 not only at 3, 4 and 10. I think big differences in 5 out of 14 molecules can’t lead the conclusion as “general trend … quite similar”

12.  Line 210, the author indicated that the barley enzyme falls into leucine aminopeptidase family. It is important to perform, i.e., phylogenetic analysis of other aminopeptidase families together with leucine aminopeptidase family including barley AP.

13.  Line 209, 210, “Our studies, although…….”. Evidences should be summarized to support this statement. Otherwise It is not clear why you can lead to this conclusion.

14.  Fig. 1, 6. Should lines of Ki vs Gibbs energy in Fig. 1 (blue line) and Fig. 6 (green line) be same. Or the Ki values in Fig. 6 (green line) are drive from different set of data than those in Fig. 1 (blue line)?

15.  Line 212, it is not clear why the authors selected the compounds 7, 8, 9, 10 and 11. Please see also major points 3 and 6.

Minor points

1.     Please carefully proof-read and spell check to eliminate grammatical errors.

2.     Line 60, “unactive” to “inactive”

3.     Line 67, please explain what “competitive inhibitors” mean. what do those molecules compete with?

4.     Line 68, lack of supporting data or reference.

5.     Fig. 1, it needs better indication. Which dots indicate molecule 7, 8, 9, 10 and 11? What is color and shape (square or circle) codes?

6.     Line 92, please cite a reference.

7.     Line 115, “Phosphoric” should be “Phosphonic”?

8.     Line 137, 139, It would be better show different clans for M1 and M17 in, i.e., phylogenetic tree.

9.     Line 140, please explain more about “similar activity-relationship”. What kind of similarity do the enzymes show towards the molecules.

10.  Line 146, please check the name of compound 11. It should be amino(3-fluoro-5-trifluoromethylphenyl)methylphosphonic acid

11.  Line 151, “izomer” should be “isomer”

12.  Line 161, “As shown earlier” please mention where.

13.  Line 166, lack of reference.

14.  Line 201, please mention which data is associated with this statement.

15.  line 209, “peliminary” to “preliminary”

16.  line 211, shouldn’t “blLAP” be “biLAP”?

17.  line 217, “vakues” to “values”

18.  line 355, “laminopeptidase” should be “aminopeptidase”